

# The role of mast cells in allergic rhinitis

Jin Zhang[1], Xiaofei Xie[2], Ruixia Ma[3] and Peng Liu[1,2,3]

[1] Department of Otolaryngology, Yibin Second People's Hospital, Yibin, China
[2] Department of Paediatric, Ziyang Traditional Chinese Medicine Hospital, Ziyang, China
[3] Second Clinical Medical College, Ningxia Medical College, Yin Chuan, Ningxia, China

Corresponding authors
Ruixia Ma, maruixia115@163.com
Peng Liu, liupengxx1213@163.com

## ABSTRACT

**Introduction**. In recent decades, mast cells and their mediators have been increasingly recognized as central players in the pathogenesis of allergic rhinitis (AR), a complex chronic nasal disease characterized by pathological changes influenced by genetic factors, various immune cells, and environmental exposures. Mast cells are pivotal in allergic reactions, orchestrating inflammation and airway contraction through the secretion of diverse mediators. Prominent among these mediators are histamine and bioactive lipids, whose physiological effects are prominently observed during the acute phase of allergic reactions. The accumulation of mast cells in specific areas of allergic rhinitis may correlate with the disease's phenotype, progression, and severity. *In vivo* experiments in mice have demonstrated that mast cells develop from mast cell progenitor cells, which are induced by inflammatory stimuli and subsequently migrate to the airway. Human mast cell progenitor cells have been identified in the bloodstream, with a high proportion potentially reflecting the persistent pathological changes associated with allergic rhinitis. The primary activation of mast cells in allergic rhinitis occurs via the cross-linking of IgE high-affinity receptors (Fc$\varepsilon$ RI) mediated by IgE in conjunction with allergens. However, mast cells can also be activated by a variety of other stimuli, including toll-like receptors and MAS-related G protein-coupled receptor X2.

**Rationale for this review**. Despite the substantial progress in understanding the role of mast cells in allergic rhinitis, several critical gaps remain in our knowledge. The complex interplay between mast cells, their mediators, and the immune system in the context of AR is still not fully elucidated. Moreover, the specific mechanisms underlying the recruitment and activation of mast cell progenitor cells in the nasal mucosa remain poorly understood. Addressing these gaps is essential for developing more effective therapeutic strategies for allergic rhinitis. This review aims to provide a comprehensive and up-to-date synthesis of the current literature on the role and development of mast cells and their progenitor cells in allergic rhinitis, including the activation pathways implicated in the pathogenesis.

**Target audience**. This review is intended for a broad audience, including researchers in the fields of immunology, allergy, and respiratory medicine, as well as clinicians who manage patients with allergic rhinitis. By summarizing the latest findings and highlighting the unresolved questions, this review aims to serve as a valuable reference for future research directions in mast cells and allergic rhinitis, ultimately contributing to improved patient care and outcomes.

## SURVEY METHODOLOGY

To ensure the comprehensiveness, completeness, accuracy, and cutting-edge nature of this review, we conducted a thorough search of the literature on mast cells and allergic rhinitis. Articles were sourced from PubMed and China National Knowledge Infrastructure (CNKI), with a focus on recent studies published within the last decade to capture the most up-to-date research findings. We employed a combination of keywords, including "mast cells," "allergic rhinitis," "mast cell progenitor cells," "activation pathways," and "pathogenesis," to identify relevant articles, and there was no time limit. The search was conducted in both English and Chinese to encompass a broader range of research. The retrieved articles were carefully analyzed, with a particular emphasis on their data and conclusions. We critically evaluated the quality and relevance of each study, ensuring that only those meeting high scientific standards were included in this review. Additionally, we cross-referenced the selected articles to identify any gaps or inconsistencies in the existing literature, thereby providing a balanced and comprehensive overview of the current state of knowledge in this field.

## INTRODUCTION

Allergic rhinitis (AR) is a common allergic inflammatory rhinopathy. With the development of the global economy and industry, as well as changes in living environments, the incidence of AR is increasing annually and has affected approximately 20% to 30% of the adult population and up to 40% of children worldwide. AR is typically characterized by persistent rhinorrhea, nasal congestion, sneezing, and/or itching, which can significantly impact patients' quality of life and lead to poor academic and work performance. AR is not only a medical issue but also a social problem that urgently requires in-depth investigation and has become a major challenge faced globally (*Hox et al., 2020*; *Bousquet et al., 2020*). The pathogenesis of AR begins when a sensitized individual is exposed to an inhaled allergen. The allergen cross-links with specific immunoglobulin E (IgE) antibodies bound to the nasal mucosal surface, thereby activating mast cells (MCs) in the nasal mucosa. This activation leads to the release of MC mediators, which initiate a cascade reaction that results in AR symptoms and facilitates the recruitment and infiltration of immune cells into the site of inflammation (*Di Lorenzo et al., 1997*; *Greiner et al., 2011*). Therefore, the activation and degranulation of MCs are crucial to the development of AR. Mast cells, first isolated from blood samples and connective tissue in 1878, have long been recognized as key effector cells in allergic reactions such as AR, allergic conjunctivitis, asthma induced by allergic reactions, urticaria, atopic dermatitis, and other diseases, playing an important regulatory role in the immune response. This review focuses on the role of mast cells in allergic inflammation and the pathogenesis of AR.

### Overview and function of mast cells

MCs are critical effector cells of the innate immune system, playing a pivotal role in the immune response against bacteria, parasites, and toxic substances. These versatile cells are widely distributed across various organs, including the skin, mucosal surfaces,

and connective tissues, where they serve as sentinels for pathogen invasion and tissue damage. Their strategic positioning allows MCs to rapidly respond to a diverse array of stimuli, releasing a broad spectrum of preformed and newly synthesized mediators that orchestrate both local and systemic immune responses. Notably, MCs possess the unique ability to interact with the nervous system, further expanding their capacity to modulate immune reactions in a highly coordinated manner (*Chia et al., 2023*). However, MCs are perhaps best recognized for their detrimental role in allergic reactions, anaphylaxis, and atopic diseases. Their degranulation in response to allergen exposure releases histamine and other inflammatory mediators, leading to the hallmark symptoms of allergies, such as bronchoconstriction, vasodilation, and increased vascular permeability. This pro-inflammatory function of MCs has long been a focus of research in the context of adult allergic diseases (*vander Elst et al., 2023*). More recently, studies predominantly in murine models have highlighted the early origins of MCs during fetal development and their potential contributions to early immune function, including the initiation of allergic responses (*St John, Rathore & Ginhoux, 2023*). These findings suggest that the foundation for allergic susceptibility may be laid during critical developmental windows, with MCs playing a key role in shaping the immune landscape from the earliest stages of life. This emerging understanding underscores the importance of MCs not only in acute allergic events but also in the long-term programming of immune responses, potentially offering new insights into the prevention and treatment of allergic diseases.

MCs are immune cells of hematopoietic origin that primarily differentiate and mature in peripheral tissues, especially those with close contact with the external environment, such as the skin, gastrointestinal tract, and airways (*Pahima & Dwyer, 2025*; *Valent et al., 2020*). Under homeostatic conditions, MCs are present in low numbers, but they can proliferate and degranulate upon stimulation, triggering allergic reactions. Traditionally, MC degranulation is initiated by the binding of IgE to the high-affinity receptor Fc$\varepsilon$RI. However, recent studies have shown that other receptors are also involved in IgE-independent degranulation. These receptors are not universally expressed in all MCs but exhibit tissue-specific patterns, allowing MCs to be classified based on receptor expression. MCs represent a highly heterogeneous cell population, with their heterogeneity largely determined by specific microenvironmental factors in peripheral tissues (*West & Bulfone-Paus, 2022*; *Yang et al., 2023*). MCs can be activated by a variety of stimuli, including IgE, complement products, viruses, bacterial particles, and endogenous peptides. Once activated, MCs release a wide range of mediators, such as histamine, proteases, cytokines, and chemokines (*Ogulur et al., 2025*). These mediators play a crucial role in innate immune responses and promote adaptive immune responses, including defense against infections and parasites, tumor suppression, and tissue homeostasis. However, impaired MC function can lead to tissue damage and a variety of pathological states, including allergies, autoimmune diseases, and tumors (*Molfetta & Paolini, 2023*; *Lecce et al., 2020*). Therefore, MCs have a dual role in immune defense and disease development.

# THE ROLE OF MAST CELLS IN ALLERGIC INFLAMMATION AND ALLERGIC RHINITIS

## Role of mast cells in allergic inflammation

The early stage of allergic reactions is mediated by MCs releasing inflammatory mediators through the production of granules, while the late stage is characterized by the influx of inflammatory cells, which are mainly regulated by MC and its secreted mediators (*Fokkens et al., 2020*; *Gelardi et al., 2022*). MCs is located on the main body surfaces exposed to the external environment, including the skin, respiratory tract, and gastrointestinal epithelium, where connective tissue MCs is located. The two subtypes of MCs can produce mucosal MCs that produce trypsin like enzymes and connective tissue MCs that produce trypsin like proteins, chymotrypsin, and carboxypeptidase. They respond faster to invasive stimuli than other tissues' intrinsic immune cells, and therefore play an important role as the host's first-line defense system against invading organisms (*Mihlan et al., 2024*; *Kim et al., 2020*). FcεRI is a high affinity IgE receptor found on the surface of MCs and eosinophils. The allergen cross-linking of FcεRI receptors bound to IgE leads to the phased release of pre formed and newly synthesized mediators. After activation, MCs degranulates, releasing storage contents of pre formed particles containing histamine, heparin, trypsin like enzymes, TNF-α, *etc.* (*Mihlan et al., 2024*; *Galli, Gaudenzio & Tsai, 2020*; *Katsoulis-Dimitriou et al., 2020*; *Metcalfe et al., 2016*; *Watts et al., 2019*; *Li et al., 2022*; *Gangwar et al., 2021*; *Zoabi et al., 2021*) (Fig. 1). MCs can affect the induction and function of adaptive immune responses, regulate histamine secretion to increase vascular permeability,and help recruit immune adaptive cells to the site of inflammation.

## The role of mast cells in the entire AR process

In the nasal mucosa, antigen-presenting cells (APCs) capture and process inhaled allergens and present them to immature Th0 cells in draining lymph nodes. Under the influence of IL-4, Th0 cells differentiate into Th2 cells, which, together with ILC2, release IL-4, IL-5, and IL-13, driving adaptive immune responses. IL-4 and IL-13 further promote B cell differentiation into plasma cells, producing specific antibodies (*Watts et al., 2019*; *Liu et al., 2022*). MCs are key effector cells in allergic reactions. Upon allergen exposure, MCs rapidly release preformed histamine, triggering acute allergic reactions that typically occur within minutes and last for 1–2 h. Histamine exerts its effects by binding to four G protein-coupled receptors (H1R-H4R) in the nasal mucosa. Specifically, H1R activation stimulates sensory nerves, transmitting signals to the central nervous system to induce itching and sneezing. Meanwhile, histamine stimulates mucus gland secretion (rhinorrhea) *via* H1R and H2R, increases vascular permeability and vasodilation, leading to nasal congestion and enhanced leukocyte recruitment (*Watts et al., 2019*; *Chen et al., 2022*; *Tatarkiewicz et al., 2019*). Following the early-phase reaction, MCs further release cytokines and chemokines, attracting inflammatory cells such as neutrophils, eosinophils, ILC2, and Th2 cells to infiltrate the nasal mucosa, forming the late-phase reaction. The late-phase reaction typically occurs 5 h after allergen exposure and lasts for 24 h. It is characterized by the recruitment of various inflammatory cells, which interact to secrete a large number of cytokines and chemokines, sustaining and prolonging the inflammatory response (*Liu et*

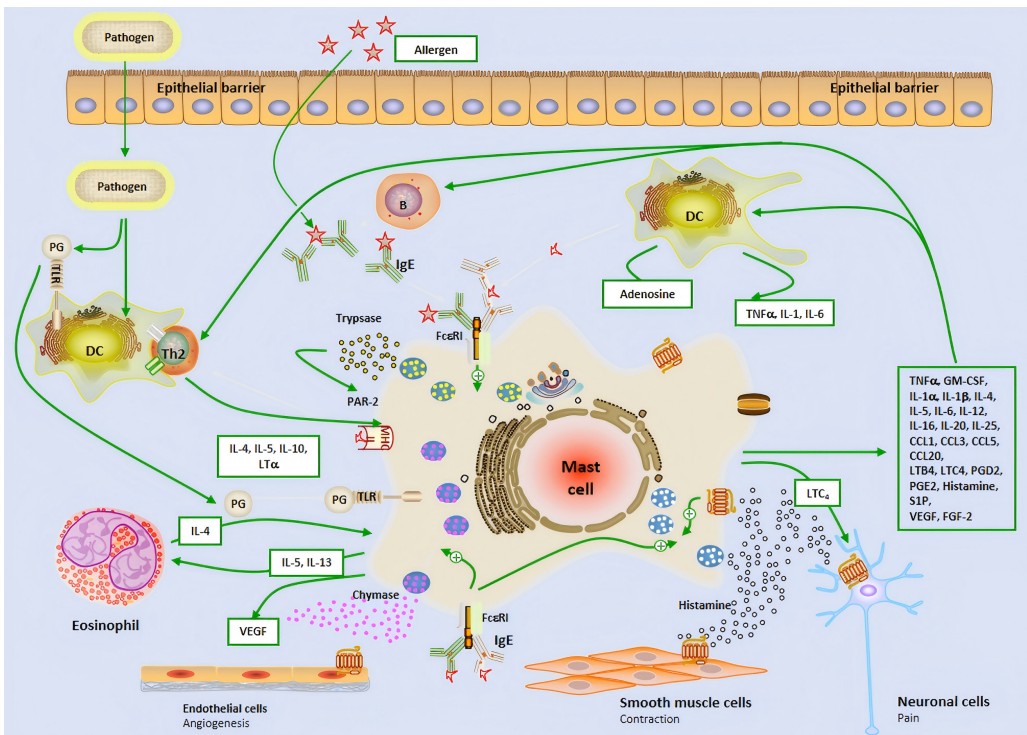

**Figure 1** **The brief mechanism of the role of mast cell in allergic airway inflammation and Allergic rhinitis.** Dendritic cell; B, B cell; P, Plasma cell; IL, Interleukin; IgE, Immunoglobulin E; Th2, Type 2 helper T lymphocyte; PG, Prostaglandin; TLR, Toll-like receptor; VEGF, Vascular endothelial-derived growth factor; PAR-2, P rotease-activated receptor 2; MHC-II, Major histocompatibility complex class II; TNF-α, Tumor necrosis factor-α; GM-CSF, Granulocyte-macrophage colony-stimulating factor; LTC4, Leukotriene C4; LTB4, Leukotriene B4; CCL, Chemokine ligand; S1P, Sphingosine-1-phosphate; FGF-2, Fibroblast growth factor-2.

*al., 2022*). Additionally, the release of cysteinyl leukotrienes and PGD2 by MCs further increases vascular permeability and promotes the recruitment and activation of ILC2 cells, thereby exacerbating the release of Th2-type cytokines and continuously driving the inflammatory response (*Thangam et al., 2018*; *Wu et al., 2016*; *Lee, Lee & Kim, 2020*) (Fig. 1). This complex cascade ultimately leads to the chronic maintenance of nasal mucosal inflammation and the exacerbation of allergic symptoms.

## MECHANISM OF ACTION OF MAST CELL ACTIVATION IN ALLERGIC RHINITIS

### IgE-dependent immunological mechanisms

When the nasal mucosa first comes into contact with a foreign allergen, it is uptaken and processed by APCs such as dendritic cells and macrophages, binds to MHCII molecules, and is then recognized by T lymphocytes, which recognize the allergen and activate to initiate a specific immune response and selectively induce specific B lymphocytes to produce specific IgE antibodies (*Bousquet et al., 2020*; *Bernstein et al., 2024*). At the same time, mast

cells have a large number of high-affinity IgE receptors (FcεRI) on the surface. FcεRI is expressed on the surface of mast cells and basophils and is a tetrameric structure composed of four chains: a α chain, a β chain, and a dimer γ chain, in which α chain constitutes an extracellular component that provides an IgE binding site, so that the body is in a state of sensitization to the antigen. When the same antigen enters the body again, by specifically binding to two or more adjacent IgE antibodies on the surface of sensitized mast cells, the membrane surface receptor FcεRI is crosslinked and activated, and the β chain and γ chain constitute intracellular components. Due to the presence of subunit immune receptor tyrosine activating motifs (ITAMs) in the β chain and γ chain, phosphorylated ITAMs play an important role in the amplification and conduction of allergic reaction signals, thereby promoting mast cell degranulation to release pre-synthesized inflammatory transmitters and mediating rapid allergic reactions. Among them, the phosphorylation process of ITAMs involves Src family non-receptor proteins, tyrosine kinases (PTKs), Lyn, Syk, and Fyn (*Peng & Zhu, 2018*; *Méndez-Enríquez & Hallgren, 2019*).

### Lyn-Syk dependent signaling pathways

The tyrosine residues on ITAMs are phosphate receptors, and after crosslinking and activation of FcεRI, phosphorylation is phosphorylated by interacting with receptor-associated tyrosine protein kinases (such as Lyn and Syk) by β and γ chains, where the γ chain plays an important role in initiating the downstream signaling pathway in the cell and the β chain plays a signal amplification role for allergic reactions (*Kanagy et al., 2022*). Lyn and Syk are both Src family non-receptor tyrosine protein kinases, of which Lyn is the first tyrosine kinase to be activated after FcεRI crosslinking activation, and the activated Lyn directly phosphorylates FcεRIβ chain and γ-chain ITAMs, and the phosphorylated FcεRI β chain can continue to recruit additional Lyn to further activate adjacent FcεRI, thereby amplifying inflammatory signals. Moreover, the ITAMs of FcεRIγ chain can provide the binding site of the tyrosine protein kinase Syk with high affinity and activate Syk, and the activated Syk further activates the transmembrane adaptor LAT and phospholipid Cγ (PLCγ), which are important in the downstream signaling pathways that regulate mast cell degranulation and release of various pro-inflammatory transmitters, such as P38, AMPK and other signaling pathways, which play an important role in the activation of mast cells and the release of inflammatory transmitters by degranulation to induce allergic responses (*Lin et al., 2016*; *Lee et al., 2013*; *Park et al., 2021*). The use of specific inhibitors and RNAi interference technology to silence Lyn found that the expression of Lyn gene in RBL-2H3 cells in the treatment group was significantly reduced, and Syk, AMPK (Ser485/491), and Akt phosphorylation were significantly inhibited, which can indicate that FcεRI activation induces AMPK (Ser485/491). Akt phosphorylation inactivation through the activation of the Lyn-Syk pathway occurs when AMPK is activated by AIC-AR, A76966, and metformin. FcεRI-mediated phosphorylation of Syk, Akt, ERK, JNK, and p38 and the release of TNF-α were inhibited, and when AMPK was inhibited by complex C, FcεRI-mediated activation of Lyn could be increased, indicating that AMPK played a negative regulatory effect on FcεRI. Based on *in vitro* experiments, they further used a mouse model of allergic dermatitis for *in vivo* experiments and stimulated it with DNP. It

was found that mice in the non-metformin treatment group induced keratinous allergies, such as increased vascular permeability, Evans blue extravasation, and swelling of the external ear, in the ear injected with DNP-IgE antibody, while the allergy symptoms were significantly milder than those in the metformin treatment group. Another study found that tussilagone (TSL) could inhibit phosphorylation of Lyn, Syk, Akt, NF-κB p65, ERK, and p38 MAPK in IgE-stimulated RBL-2H3 cells, improve rhinitis symptoms, reduce nasal mucosal pathological changes, reduce the production of IgE, histamine, and IL-6, and have a protective effect against OVA-induced allergic guinea pigs. The results show that one of the mechanisms of TSL's anti-AR effect is the inhibition of the Lyn/Syk, NF-κB, and p38 MAPK signaling pathways in activated mast cells (*Jin et al., 2020*). *Wang et al. (2022)* found that alpha-linolenic acid (ALA) inhibits IgE-mediated $Ca^{2+}$ mobilization, degranulation, and cytokine release in allergic disease laboratory 2 (LAD2) cells, and Western blot results showed that ALA downregulated the FcεRI/Lyn/Syk signaling pathway by inhibiting Lyn kinase activity. These all indicate that Lyn-Syk can regulate AR by regulating downstream signaling pathways, which will provide a new target for future treatment of AR.

### Fyn-dependent signaling pathways

Fyn also belongs to a kind of Src family non-receptor tyrosine protein kinase, which also plays a key role in the process of FcεRI crosslinking activation inducing mast cell degranulation, after FcεRI crosslinking activation, Fyn is activated, and then cytosolic linker molecule GAB2 phosphorylation activation, activated GAB2 binds to phosphatidylinositol 3 kinase (PI3K), resulting in enhanced $Ca^{2+}$ mobilization, triggering degranulation to release histamine, prostaglandins and leukotrienes, etc., causing smooth muscle contraction and gland hypersecretion, increased vascular permeability, mucosal epithelial swelling, that is, early phase reaction, and prostaglandins, leukotrienes can also recruit and activate inflammatory cells in the nasal mucosa, aggravate nasal mucosal inflammation and hyperresponsiveness, that is, delayed phase reaction (*Lee et al., 2013*; *Kim et al., 2015*; *Yoo et al., 2014*). Another study found that dasatinib inhibits the activation of Syk and Syk-mediated downstream signaling proteins LAT, PLCγ1, and three typical MAP kinases (Erk1/2, JNK, and p38), which are crucial for the activation of mast cells. In vitro tyrosine kinase assays demonstrated that dasatinib directly inhibits the activity of Lyn and Fyn, tyrosine kinases upstream of Syk in mast cells. This indicates that dasatinib can inhibit Lyn and Fyn, members of the Src family of kinases, and suppress mast cells and passive cutaneous anaphylaxis (PCA) both *in vitro* and *in vivo*. Therefore, the possibility of repositioning dasatinib for the treatment of various mast cell-mediated allergic diseases has been highlighted (*Lee et al., 2020*).

## Non-immunological mechanisms

Among the activation and degranulation factors of AR-induced mast cells, the IgE-dependent immunological mechanism is the main activation pathway, but in addition, there are many other receptors on the surface of mast cells, such as c-Kit receptors, complement receptors (C3aR, C5aR), Toll-like receptors (TLRs), chemokine receptors (CCR3), *etc.*, which can also stimulate mast cell activation and degranulation by binding to the corresponding ligands (*Metcalfe, Baram & Mekori, 1997*).

### C-Kit receptor-dependent signaling pathways

In mast cells, C-Kit remains highly expressed during development, and C-Kit signaling is critical for mast cell development (*Maeda et al., 2010*). C-Kit (CD117) is a receptor with tyrosine kinase activity on the surface of mast cells that exists throughout the life cycle of mast cells. The growth, survival, differentiation, and homing of mast cells mainly rely on the activation of the C-KIT receptor (CD117). Stem cell factor (SCF) is a ligand of the c-Kit receptor. SCF binds to the c-Kit receptor, causing c-Kit receptor phosphorylation. The C-Kit receptor can activate several signaling pathways. For example, PI3K and mitogen activate protein kinase pathways, thereby promoting mast cell degranulation and releasing inflammatory mediators (*Ribatti, 2016*; *Tsai, Valent & Galli, 2022*). *Wu et al. (2012)* and *Wu et al. (2014)* used nasal topical administration and systemic administration of c-kit siRNA in allergic airway model mice to interfere with the expression of c-Kit and found that the expression of c-Kit decreased in both modes of administration, the inflammatory cell infiltration decreased, the production of Th2 cytokines, IL-4 and IL-5, was reduced, and airway symptoms were significantly reduced, indicating that c-Kit receptors played an important role in inducing allergic reactions induced by mast cell activation.

### Complement receptor-dependent signaling pathways

The complement system is an integral part of the body's innate immune system that can recognize threats from foreign and intrinsic pathogens, and in recent years, a large number of studies have found that complement system regulation is associated with the onset of a variety of diseases. Among them, the synergistic activity between mast cells and the complement system plays a key role in many allergic, skin, and vascular diseases, and mast cells, as important effector cells of AR, can express receptors (C3aR, C5aR) of complement allergic toxins C3a and C5a on the surface of mast cells (*Komi et al., 2020b*). A large number of studies have found that C3a and C5a always occur during IgE-mediated rapid allergic reactions, which may be through the body's three pathways activated by complement (the classical pathway, the lectin pathway, and the alternative pathway) to produce the C3 converting enzyme. C3 conversion enzyme acts on C3 to cleave it to produce complement fragments C3a and C3b, and then C3b can further act to generate C5 converting enzyme and act on C5 to produce complement fragments C5a and C5b, C3a and C5a bind to mast cell surface receptors. Activation of mast cells by recruiting G protein subtypes leads to increased $Ca2^+$ concentrations in mast cells, leading to activation of mitogen-activated protein kinase, PI3K, and PKB/AKt pathways that stimulate mast cell degranulation (*Laumonnier, Schmudde & Köhl, 2011*; *Schäfer et al., 2013*; *Yanase et al., 2021*). *Li, Xu & He (2015)* found a significant decrease in serum C3a and C5a levels after three years of desensitization therapy when using sublingual immunotherapy in AR patients, but they were unable to detect any correlation between symptomatic drug scores and C3a and C5a levels during SLIT.

### Toll-like receptor-dependent signaling pathways

Toll-like receptors (TLRs) are important components of innate immunity; they are recognition receptors expressed on the surface of intestinal mucosal lymphocytes and epithelial cells, provide a defense barrier inflammatory response to invading pathogens,

are located in the cytoplasmic membrane and also in the endosome, and can detect a range of pathogen-related molecular patterns in bacteria, viruses, and fungi (*Kirtland et al., 2020*). TLRs are molecular pattern recognition receptors associated with pathogen recognition that play an important role in both innate and acquired host immunity, and most studies have found that TLRs are mainly expressed in a variety of immune cells, such as dendritic cells and mast cells (*Zhang et al., 2016*; *Numata, Harada & Nakae, 2022*). So far, the study has found that 11 members of the TLR family have been recognized, in which TLR1, TLR2, TLR4, TLR5, TLR6, TLR10, and TLR11 are expressed on the cell surface to recognize proteins, lipoproteins, and polysaccharides in bacteria, while TLR3, TLR7, TLR8, and TLR9 are located in intracellular vesicles and sense different types of microbial nucleic acids. Microorganisms can act on the surface of mast cells through TLR2 and TLR4 to directly activate mast cells. Mast cells can react with peptidoglycan (PGN) and lipopolysaccharides (LPS) through TLR2 and TLR4, activate the signaling pathway MyD88 and β interferon TIR domain linker (TRIF), and finally activate NF-κB. Under the stimulation of PGN, mast cells secrete important cytokines that promote allergic inflammation, such as IL-4, IL-5, TNF-α, etc. These cytokines play an important role in the pathogenesis of AR (*Liu & Tao, 2012*). At the same time, *Cui et al. (2015)* found that compared with the control group, the expression of TLR2 and TLR4 in persistent AR was significantly higher, and the expression of IL-6 and IL-8 was also significantly higher, which indicated that TLR2 and TLR4 may be one of the main reasons for the persistence and aggravation of allergic inflammation.

### Chemotokine receptor-dependent signaling pathways

The chemokine receptor CCR3 is a seven-transmembrane G protein-coupled receptor (GPCR) that plays a crucial role in allergic inflammation. Initially, CCR3 was believed to be specifically expressed in eosinophils, but subsequent studies revealed its presence in Th2 lymphocytes, basophils, and mast cells, upon ligand binding, CCR3 promotes the infiltration of inflammatory cells into tissues, contributing to allergic responses (*Shao et al., 2022*; *Komi et al., 2020a*). In mast cells, CCR3 activation triggers G protein-mediated signaling, releasing the FcεRIα subunit and coupling to PI3Kγ, thereby amplifying degranulation *via* Ca2$^+$ influx. *In vitro* experiments have shown that inhibiting CCR3 expression significantly reduces allergen- and IgE-mediated mast cell degranulation (*Miyazaki et al., 2009*). In a murine model of allergic rhinitis, *Wu et al. (2020)* demonstrated that CCR3 shRNA effectively downregulated CCR3 expression in bone marrow, peripheral blood, and nasal mucosa. This intervention reduced nasal symptoms, OVA-specific IgE levels, and the infiltration of inflammatory cells and mast cells into the nasal cavity, while also alleviating histopathological changes in the nasal mucosa. Additionally, CCR3 shRNA treatment significantly decreased levels of histamine, tryptase, and PGD2 in both peripheral blood and nasal mucosa. These findings highlight that shRNA-mediated CCR3 inhibition effectively attenuates mast cell migration, infiltration, and degranulation in local tissues, thereby reducing inflammation in allergic rhinitis mice. Furthermore, studies have shown that CCR3 deficiency in mice impairs eosinophilic

inflammation and cytokine release, leading to a shift from Th2-mediated eosinophil-driven asthma to an innate-like mast cell and neutrophil-mediated lung inflammation. This suggests that CCR3 plays a pivotal role in both innate and adaptive immune responses during allergic inflammation. Collectively, these findings underscore the potential of targeting CCR3 for therapeutic intervention in allergic diseases.

## SUMMARY AND OUTLOOK

In summary, AR is a heterogeneous disease, with variations in onset time, severity, inflammatory patterns, and responsiveness to corticosteroid treatment. Similarly, we believe that mast cells play either a primary or secondary role in the pathogenesis of AR. Given that mast cells can secrete a wide range of mediators, the beneficial effects of histamine receptor antagonists are limited. Currently, mast cells are considered the main effector cells and immune regulatory cells in AR, playing a significant role in its pathogenesis. Further in-depth research has revealed that their related mechanisms of action still need to be further elucidated. This article reviews the recent literature on the role of mast cells in allergic airway diseases and finds that mast cells are strictly regulated at various stages of differentiation, maturation, activation, and degranulation. Their mechanisms of action in AR involve many relevant signaling pathways.

**Abbreviations**

| | |
|---|---|
| **DC** | dendritic cell |
| **B** | B cell |
| **P** | Plasma cell |
| **IL** | Interleukin |
| **IgE** | Immunoglobulin E |
| **Th2** | Type 2 helper T lymphocyte |
| **PG** | Prostaglandin |
| **TLR** | Toll-like receptor |
| **VEGF** | Vascular endothelial-derived growth factor |
| **PAR-2** | Protease-activated receptor 2 |
| **MHC-II** | Major histocompatibility complex class II |
| **TNF-α** | Tumor necrosis factor-α |
| **GM-CSF** | Granulocyte-macrophage colony-stimulating factor |
| **LTC4** | Leukotriene C4 |
| **LTB4** | Leukotriene B4 |
| **CCL** | Chemokine ligand |
| **S1P** | Sphingosine-1-phosphate |
| **FGF-2** | Fibroblast growth factor-2 |

### Funding

The authors received no funding for this work.

## Competing Interests

The authors declare there are no competing interests.

## Author Contributions

- Jin Zhang performed the experiments, authored or reviewed drafts of the article, and approved the final draft.
- Xiaofei Xie performed the experiments, prepared figures and/or tables, and approved the final draft.
- Ruixia Ma conceived and designed the experiments, authored or reviewed drafts of the article, and approved the final draft.
- Peng Liu conceived and designed the experiments, analyzed the data, prepared figures and/or tables, and approved the final draft.

## Data Availability

This is a literature review.

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
