# Peer review of "The role of mast cells in allergic rhinitis"

_PeerJ, doi:10.7717/peerj.19734_

## Round 0.1 · original submission · Minor Revisions

As recommended by the reviewers, the article requires minor revisions.

**Language Note:** PeerJ staff have identified that the English language needs to be improved. When you prepare your next revision, please either (i) have a colleague who is proficient in English and familiar with the subject matter review your manuscript, or (ii) contact a professional editing service to review your manuscript. PeerJ can provide language editing services - you can contact us at [email protected] for pricing (be sure to provide your manuscript number and title). – PeerJ Staff

Reviewer 1 ·

Basic reporting

Language and Clarity: The manuscript is generally understandable.
Structure and References: The structure is logical and flows well from introduction to conclusion. The references are abundant and reflect recent research.
Background and Motivation: The rationale for this review is clearly stated, and the intended audience is well defined.

Experimental design

Methodology Description: The literature search methodology is clearly described, using both PubMed and CNKI, and includes English and Chinese sources, which is commendable.
Comprehensiveness and Balance: The review focuses on recent studies and aims to be comprehensive.

Validity of the findings

Conclusion Appropriateness: The conclusions are consistent with the content and appropriately highlight unresolved questions and future directions.
Logical Flow and Support: The discussion of mast cell activation pathways (both IgE-dependent and independent) is detailed and supported by scientific evidence.

Additional comments

Strengths
1. The review provides a thorough summary of recent findings on mast cells, especially regarding signaling pathways, which will be valuable to specialists.
2. Figure 1 is a helpful visual aid that enhances understanding.

Areas for Improvement
1. Author described “3.1 IgE-dependent immunological mechanisms” and “3.2 Non-immunological mechanism” as opposite expression. But, I think that 3.2 Non-IgE-dependent immunological mechanisms is better.
2. This review also targets clinicians. Therefore, I believe it would be beneficial to include current treatment options targeting mast cells, as well as potential future therapies, within the review.

Cite this review as

·

Basic reporting

I find this article a useful compendium on the state of the art on research in mastocyte pathway in rhinitis. The original figure can be useful because there are a lot similar involving Il5 and th2 pathways but there is a lack in Mastocytes.
Language is clear and phrases consequential.
The only flaw I can find is in lack of clarity in article selection and presentation of the consequential findings.
I see no signs of heavy AI use in the manuscript

Experimental design

I have some perplexities about clarity in the methods.
RR- 68-70 It could be useful to include the query string you used and if there were time limitations ( for example: from 2020, from 1975)
RR72-73 Can you specify the high scientific standards? Can you add an example of a non-included article?
RR245-251 this is an example of the issue: all articles are treated as peer in evidence.
There are articles that conflicts one witch another? Others with higher levels of confidence? Articles with weak stanrdards or confronts with different models as murine, in vistro or human?

Validity of the findings

Findings are presented in a clear and logic structures.
In this kind of work novelty is secondary and i think that a solid explicative review can be a good foundation for study and research.
I liked the figure, clear and crafty made.

The strength of the findings is as strong as the selection methods and that is the motive that makes me ask for clarity.

Cite this review as

---

## Round 0.2 · accepted · Accept

As recommended by the reviewers, the article can be accepted.

Reviewer 1 ·

Basic reporting

No comments

Experimental design

No comments

Validity of the findings

No comments

Additional comments

I accept the artiucle.

Cite this review as

·

Basic reporting

Now it is more clear!

Experimental design

the study design was already good, now is more clear how they did it

Validity of the findings

The level of evidence is now clear and more understandeable for the audience, thank you.

Additional comments

I think this new version is more solid and clear, have a nice day

Cite this review as